# Evaluation of Foam Gel Compound Profile Control and Flooding Technology in Low-Permeability Reservoirs

Xiaoyu Gu [1,*], Gejun Cai [1], Xuandu Fan [1], Yanlong He [1], Feifei Huang [2], Zhendong Gao [3] and Shaofei Kang [2]

[1] School of Petroleum Engineering, Xi'an Shiyou University, Xi'an 710065, China; 15614323118@163.com (G.C.); 13355717378@163.com (X.F.); stpnet@126.com (Y.H.)
[2] School of Petroleum Engineering and Environmental Engineering, Yan'an University, Yan'an 716000, China; ffhuang@yau.edu.cn (F.H.); kangshaofei2020@163.com (S.K.)
[3] Yanchang Oilfield Co., Ltd., Yan'an 716000, China; rx.52825@163.com
* Correspondence: 190108@xsyu.edu.cn

**Abstract:** In the waterflooding development of fractured ultra-low permeability reservoirs, the heterogeneity is becoming increasingly serious. The development of large fracture channels leads to serious water channelling and low recovery, and the effect of conventional profile control is not ideal. This paper proposed gel foam composite profile control and flooding technology to solve the above problems. Herein, the new intelligent gel and foaming agent systems were optimized through laboratory experiments, and their performance was evaluated. The new intelligent gel system has the characteristics of low viscosity, easy preparation, good injection, slow cross-linking, high strength, and long-term effectiveness. The injection parameters were optimized, and the indoor injection scheme was formulated, that is, the optimal injection volumes of gel and foam slugs were 0.3 and 0.6 PV, respectively. The injection sequence of composite slugs was to inject gel slugs first, then foam slugs. The injection mode of air foam slugs was multiple rounds of small slug injection. The final recovery rate in the indoor dual tube oil displacement experiment reached 35.01%, increasing by 23.69%. Furthermore, an oil output increase of 899 t and an average water cut decrease of 5% were acquired in the oil field test. It shows that the injection scheme can effectively improve oil recovery. The gel foam compound profile control and flooding technology herein has good adaptability in similar reservoirs and has good promotion prospects.

**Keywords:** low permeability reservoir; foam; gel; compound profile control and flooding

## 1. Introduction

China is rich in ultra-low permeability oil reservoirs, which are widely distributed in Ordos, Songliao, Zhunkar, and Bohai Bay Basins [1–3]. The Yanchang Formation in Ordos Basin is a typical ultra-low permeability reservoir. Due to the low natural energy of reservoir, water injection development has become the only way to stabilise production of such reservoirs [4,5]. In the process of developing water injection, the water injection is easy to open due to the development of natural fractures in the reservoir. Some water injection wells are converted from oil wells, leading to forming water channelling channels between oil and water wells, resulting in a low utilization rate of water injection capacity, ineffective circulation, and seriously affecting the water drive development effect of water injection [6–8]. Profile control and water plugging technology have become necessary to ensure the water injection effect. However, conventional profile control agents are either "unable to inject" or "unable to block". In this case, gel foam composite profile control and flooding technology are proposed. Gel foam composite profile control and flooding combine the dual advantages of gel and air foam flooding, which can well improve the heterogeneity of the reservoir and has the dual role of oil displacement and profile control [9,10]. This is of great significance to the research on high-efficiency water injection development of low-permeability reservoirs.

Fried et al. [11] highlighted that the main mechanism of foam flooding is to reduce gas permeability and delay premature gas breakthrough. Jensen et al. [12] found through experiments that in the absence of oil in the porous medium, the flow rate has a certain influence on the foam structure, which is mainly manifested in the refinement with the increased flow rate. When oil is present, foam formation is related to oil saturation, and when oil saturation exceeds a certain level, foam formation is not possible.

Thomas et al. [13] believed that the infiltration channel of foam depends on the saturation of the liquid. Refs. [14,15] proposed injecting dilute sodium silicate alkaline solution into the formation and then injecting $CO_2$ gas to generate a gel to plug the high permeability layer. Owette et al. [16] conducted an experiment with an indoor $CO_2$ foam displacement effect, and the results showed that temperature and pressure have a certain impact on the stability of the foam. If the pressure increases, the stability of foam can increase, but the increase in temperature is not conducive to the stability of the foam. Alexandrov et al. [17] studied the clearing law of foam fluid in porous media. In the experiment a new idea was put forward: analyse the size and structure of different pore masks using statistical methods. Zitha et al. [18,19] studied the change in foam precision and the basic seepage law in porous media. Using statistical methods, the formation and bursting laws of foam in the formation have been recognized. Nguyen et al. [20–22] studied the influence of cochannelling flow in the lower layer on the foam paint flow gauge under oil-free conditions using CT technology. They found that for heterogeneous formation, whether there is an interlayer fluid exchange, the flow channel of foam fluid mainly occurs in the formation high permeability channel and then flows to the low paint layer. Bath et al. [23] conducted a simulation test study on a high-pressure light oil reduction in Beihai Oilfield. The result is that air drives can significantly improve the oil recovery of the reservoir. The simulation test shows that the oil recovery of oil leakage has increased from 37% to 48%. Chen et al. [24] studied the factors that affect foam quality, size, and stability. The research results showed that factors such as polymer concentration change, gas flow rate, and storage time had a certain impact on foam quality, size, and shape. Wang et al. [25] developed a new coagulated and dispersed foam system, which can improve the liquid efficiency and yield of subsequent water flooding. To solve the problem that the high gas–oil ratio of production wells affects normal production, Pu et al. [26] selected a micro crosslinked foam gel system to prevent gas channelling and achieved good results. Yan et al. [27] conducted a pilot test of air foam profile control and flooding for two water injection well groups in Well Block Tang 80 of Gangushi oilfield. Wang et al. [28] conducted an indoor study on air foam-enhanced oil recovery in Block Hu 12 of the crude oilfield. They established a heterogeneous reservoir model according to reservoir conditions. The displacement experiment showed that when the gas-liquid ratio was 3:1, and the air foam was injected alternately, the effect was the best, and the final oil recovery could be as high as 80.3%.

This paper evaluated the properties of the existing air foam profile control and flooding agent system in the Baota oilfield based on the water source in the study area and the water samples at the outlet of the sewage treatment station. Accordingly, the foaming agent, profile control, and flooding agent systems used in the current market were optimised and screened. In more detail, a composite profile control and flooding system suitable for the reservoir conditions in the study area was screened and optimised in order to evaluate the static performance of the optimised composite profile control and flooding system to determine its clogging capacity and to provide guidance for optimising the supporting process parameters. Consequently, a laboratory physical simulation experiment was conducted to optimise the process parameters, such as the slug sequence and the slug injection amount of the supporting working fluid, and to determine the construction process of the foam–gel composite profile control and flooding in the study area.

## 2. Materials and Methods

Distilled water was used to prepare 100 mL of different foaming agent base solutions with a concentration of 0.5%, the high-speed agitator was set at 3000 r/min, and were mixed for 3 min. Subsequently, the agitator was closed and immediately the foaming volume was read in the measuring cylinder. When 50 mL of foam liquid was separated, the required time and the half-life was read. The specific experimental methods were as follows: (1) the temperature was controlled at $30 \pm 1$ °C, the proportion of solvent and solute was calculated according to the sample concentration, different samples of foaming agent were added to clean water, formation water or sewage, and a series of foaming agent solutions were prepared. (2) A total of 100 mL of foaming agent solution was put into the mixing cup, and the speed of mixer was adjusted to 7000 rpm/min before stirring for 2 min. (3) The stirred foam was poured into a 1000 mL measuring cylinder; the foam volume (V) was read and recorded, and the stopwatch was pressed to count this time. (4) Change in foam volume was recorded with time, and the time (t) was recorded when 50 mL liquid was separated from the measuring cylinder as the half-life of foam system, and it was expressed as $t_{1/2}$. (5) Measured $V$ and $t_{1/2}$ were used in the formula: $F = V \times t_{1/2}$. The corresponding foam comprehensive index F was calculated by $t_{1/2}$, which was used as an index to evaluate the comprehensive performance of the foam liquid system.

The optimal experimental steps for injecting the gel slug were as follows: (1) Core was placed in core barrel, the test device was connected, the airtightness of pipeline and valve was checked, and confining pressure was applied. (2) A pressure test was carried out to ensure that there was no gas leakage at any point, the permeability was measured by gas, the dry weight was measured, and the core was vacuumed. (3) The core sample saturated with formation water was pumped, the amount of saturation was recorded, and the permeability and porosity of the water phase were calculated. (4) Intermediate container and saturated oil were exchanged, and the irreducible water and original oil saturations were calculated. (5) Water drive was started at the injection rate of 0.4 mL/min until the water content reached 98%, and the time; water drive oil displacement efficiency, oil displacement pressure, liquid production, and oil production were recorded. (6) A displacement test was then performed and time, displacement pressure and other parameters were recorded. Two cores of different permeability were displaced in parallel. The fractured core simulated the high permeability layer, and the sand fill model simulated the low permeability layer to simulate reservoir heterogeneity.

## 3. Results and Discussion

### 3.1. Foaming Agent Optimization

Under same concentrations (0.5%), temperatures (40 °C), pressures (1 atm.), pH (7.2), and salinity (35,000 mg/L), and the foaming properties of six foaming agent solutions were evaluated using the Warning blender method. The experimental results are shown in Table 1. Under the same experimental conditions, foam volume, foam half-life, and comprehensive foam value were combined. The results showed that HDSX-1, a foaming agent developed by self-modification, had better performance than other foaming agents. Therefore, HDSX-1 was selected as the foaming agent of system in this paper.

### 3.2. Optimization of Nanoparticle Composite Foam

Figure 1 shows the half-life of foam prepared by nano-silica particles of different hydrophobic types (N1–2, N1–5, N1–10, N1–15, and N1–18) and foaming agent HDSX-1. During the experiment, the concentration of five kinds of nanoparticles was fixed at 1.5%, and the foaming agent concentration varied from 0.05% to 1.0%. When the concentration of the foaming agent solution was lower than 0.5%, the stability of the foam prepared by five hydrophilic particles and the foaming agent gradually increased with increasing concentration of the foaming agent solution. When the concentration of foaming agent solution was higher than 0.5%, the stability of foam prepared by five hydrophilic particles and the foaming agent was unchanged with increasing concentration of foaming agent

solution. Therefore, when the concentration of the foaming agent solution was 0.5%, five hydrophilic particles could form the best synergistic foam stabilizing effect with the foaming agent. This was mainly because HDSX-1 could form monolayer adsorption on the particle surface at an appropriate concentration of foaming agent, enhancing the hydrophobicity and surface activity of particles, making the particles more easily adsorbed on the gas–liquid interface, and subsequently stabilizing the foam. Among them, the synergistic effect of N1–2 and HDSX-1 was the best, and the stability of foam prepared by nanoparticles and foaming agent HDSX-1 was significantly improved. Therefore, the foaming agents HDSX-1 and N1–2 were selected to be compounded herein.

**Table 1.** Performance evaluation results of different types of foaming agents.

| No. | Foaming Agent | Foam Volume/mL | Half Life/s | Comprehensive Value |
|-----|--------------|----------------|-------------|---------------------|
| 1 | CTAC | 360 | 217 | 78,120 |
| 2 | BZ-1 | 390 | 212 | 82,680 |
| 3 | SD-Al | 280 | 154 | 43,120 |
| 4 | SDl | 280 | 140 | 39,200 |
| 5 | BK-2 | 480 | 200 | 96,000 |
| 6 | BK-7 | 475 | 201 | 95,475 |
| 7 | No. 1 foaming agent | 530 | 400 | 21,200 |
| 8 | PO-FA330 | 650 | 348 | 226,200 |
| 9 | BK6A | 595 | 335 | 211,225 |
| 10 | HDSX-1 | 678 | 420 | 243,985 |

Note: CTAC Cetrimonium chloride; BZ-1 2-alkyl Imidazoline; SD-Al Sodium aliphatic alcohol sulfate; SD1 Sodium lauryl oxyethyl sulfonate; BK-2 Tetradecyl aminopropionic acid; BK-7: Hexadecylaminopropionic acid; No. 1 foaming agent; Foaming agent used in the Baota oil field; PO-FA330: Octyl phenol ethoxylate; BK6A Acetylene glycol polyoxyethylene ether; HDSX-1 Bio-based anionic foaming agent.

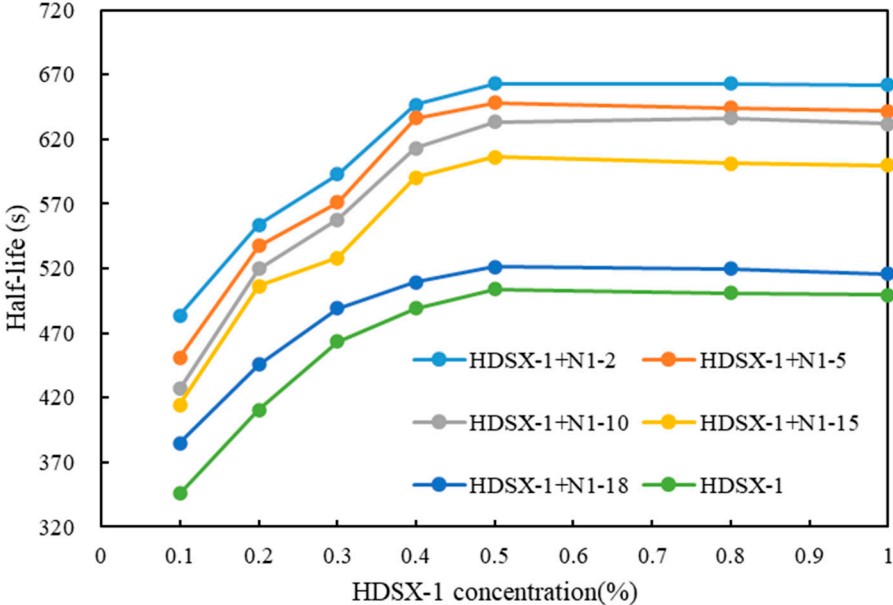

**Figure 1.** Stability of foam of $SiO_2$ and HDSX-1 composite system.

### 3.3. Salt Resistance of Different Polymer Solutions

A Brinell viscometer was used to compare the different types of polymer solution at room temperature (30) and 80 °C when the polymer concentration was 0.3%, and the selected salinity was 10,000, 20,000, 30,000, and 40,000 mg/L. Figure 2 shows the effect of salinity on the viscosity of different polymers. When deionized water was used to prepare the polymer solution, after aging for 24 h, the polymer solution showed higher viscosity, which was 1860, 1900, 1620, and 1680 mPa·s at room temperature, respectively. This was because, in deionized water, partially hydrolysed polyacrylamide molecules are

unaffected by ions, and polymer molecules could stretch as far as possible in an aqueous solution, showing high apparent viscosity. However, with an increase of salinity in the solution, inorganic salt ions continuously compress the diffusion double electron layer on the polymer molecular chain, compressing the polymer molecular coil, and significantly reducing the polymer viscosity of solution. When the salinity was 10,000 mg/L, the viscosity of different types of polymer solutions aged for 24 h at room temperature was significantly reduced to 870, 620, 555, and 1085 mPa·s, respectively. Accordingly, the salinity has a very significant impact on the viscosity of the DQ-1, DQ-2, and Mo-4000 polymers. With a further increase in salinity, the viscosity decreases. However, polymer AP-P4 was less affected by salinity. When the salinity in the solution reached 34,711.8 mg/L, the viscosity of the polymer solution aged for 24 h at room temperature was 321, 358, 440, and 980 mPa·s, respectively. Therefore, combined with the salinity of formation water in the study area (34,711.8 mg/L), the polymer AP-P4 showed a good salt resistance effect.

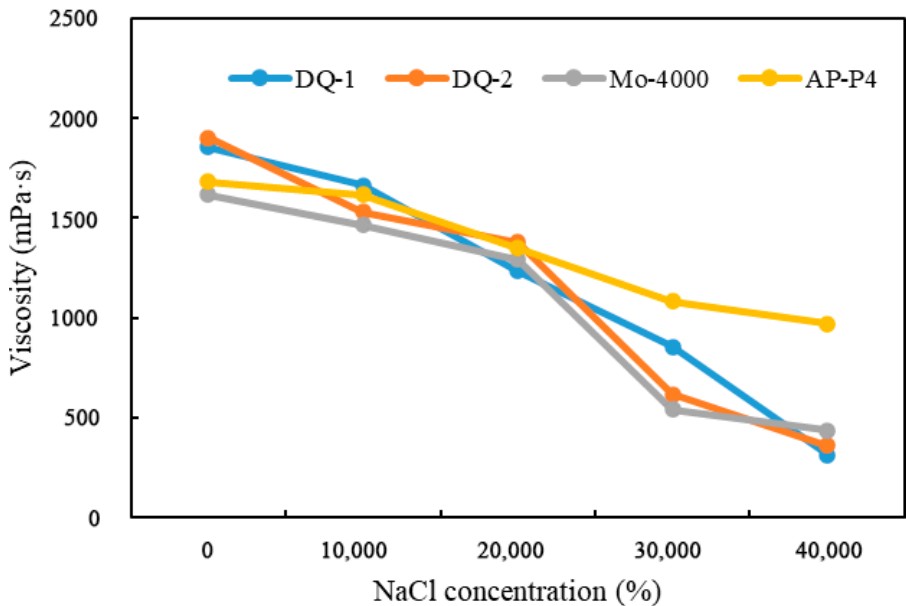

**Figure 2.** Effect of salinity on the viscosity of different polymer solutions.

*3.4. Temperature Resistance of Different Polymer Solutions*

One of the main factors affecting the gelling performance of a polymer gel system is reservoir temperature. The temperature resistance of the polymer AP-P4 solution was evaluated using a Brookfield viscometer. The polymer concentration was 0.3%, and the salinity was 34,711.8 mg/L for preparation. The aging time of the polymer solution at 30, 50, 70, and 90 °C was 24 h. Figure 3 shows the viscosity of the polymer solution at different temperatures. When the salinity in the solution reached 34,711.8 mg/L, the viscosity of the polymer solution aged at 30 °C for 24 h was 300, 328, 425, and 956 mPa·s, respectively. However, with the increasing temperature, the viscosity of the polymer solution showed a downward trend. After aging at 90 °C for 24 h, the viscosities of four polymer solutions were 15, 20, 22, and 320 mPa·s. The viscosity of the AP-P4 solution was relatively high. This shows that the additional NVP monomer added in polymer AP-P4 can increase the thermal stability of the polymer, mainly because there are rigid groups with strong thermal stability in the NVP structure. The NVP structure can form a large steric hindrance, and inhibit the hydrolysis of amide groups, so it can significantly enhance the temperature resistance of polymer solution. Therefore, combined with the temperature (30.6 °C) of Chang 6 reservoir in the study area, the polymer has a good temperature resistance effect. Therefore, AP-P4 was the polymer used in this optimized system. In summary, the key treatment agent used in the foam gel composite flooding system in this article was the

nanoparticle composite foam agent in a combination of HDSX-1 and N1–2, the polymer was AP-P4 and the crosslinking agent was polyethyleneimine PEI-4 with a good field use effect.

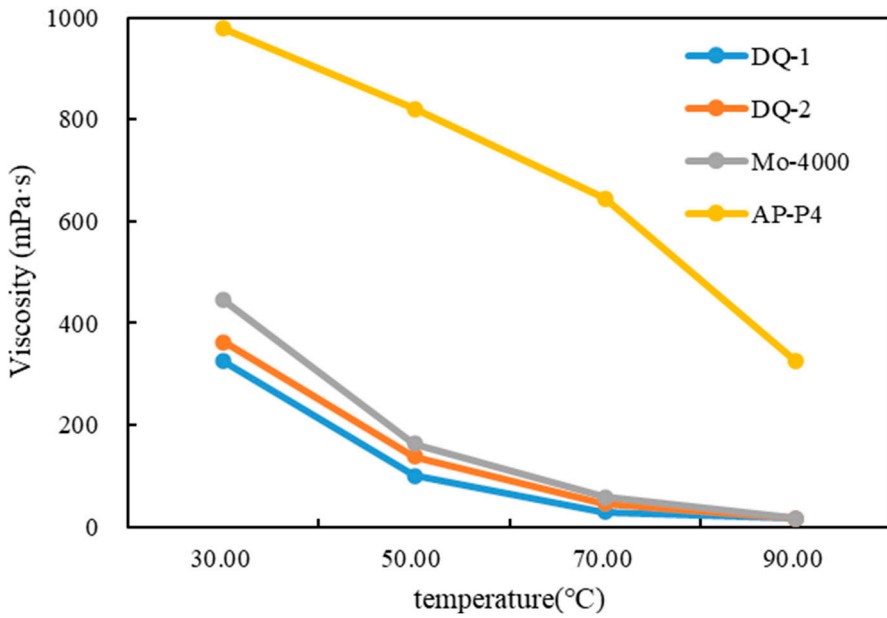

**Figure 3.** Effect of temperature on viscosity of different polymer solutions.

### 3.5. Optimizing Total Injection Volume of Gel Slug

Four fractured cores with similar properties were selected for the experiment, and Table 2 depicts the core parameters. The injection amount of gel was 0.15, 0.25, 0.3, and 0.45 PV, respectively, and the slug size of four gels was optimized indoors to optimize the injection amount of gel. Each core was first waterflooded to 98% water cut and then injected with gel. After 24 h of gel, water was injected at 0.3 PV until the water drive pressure became stable. The gel injection pressure and water drive pressure were recorded after gelling, and the resistance factor was calculated. Taking the fractured core LE-A-3 as an example, 0.3 PV of gel was injected after water flooding, water was continued to be injected after 24 h of gel, and the change of water injection pressure was observed (Figure 4). The injection pressure was slight when injected with 0.3 PV of gel. After gelling, the injection pressure rose rapidly, then increased gently, and the increase gradually decreased. The injection pressure of injected gel was low, the effect of plugging fractures was good, and it had a good adaptability to the low-permeability formation in the study area. Changes in water drive pressure of 0.15, 0.25, 0.3, and 0.45 PV gel segments after plugging the gel were plotted (Figure 5). The resistance factor was greatly improved only when the injection amount of gel was greater than 0.3 PV, meaning that only 0.3 PV gel can be injected to seal the fracture and improve the subsequent water drive sweep range. Therefore, it is recommended to inject 0.3 PV gel into the fractured formation.

**Table 2.** Plugging experimental scheme under different injection amount of gel.

| No. | Fracture Permeability/$10^{-3}$ $\mu m^2$ | Pore Volume/$cm^3$ | Fracture Pore Volume/$cm^3$ | Gel /PV | Subsequent Water Drive Volume/PV |
|---|---|---|---|---|---|
| LE-A-2 | 230 | 69.8 | 6.05 | 0.15 | 0.3 |
| LE-A-2 | 238 | 68.5 | 6.19 | 0.25 | 0.3 |
| LE-A-3 | 239 | 70.5 | 6.18 | 0.30 | 0.3 |
| LE-A-4 | 226 | 69.8 | 6.16 | 0.45 | 0.3 |

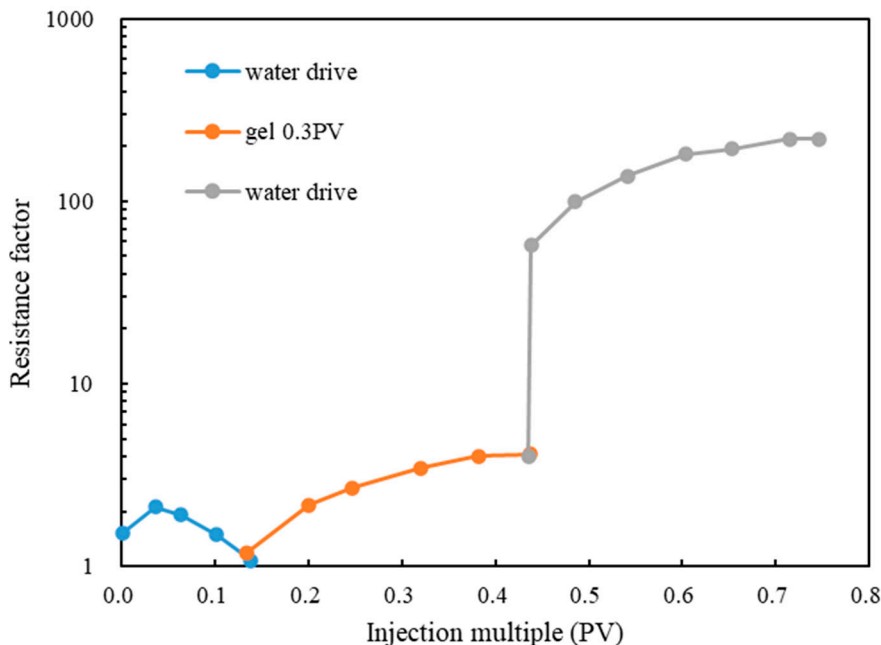

**Figure 4.** Change curve of driving resistance of 0.3 PV injected gel.

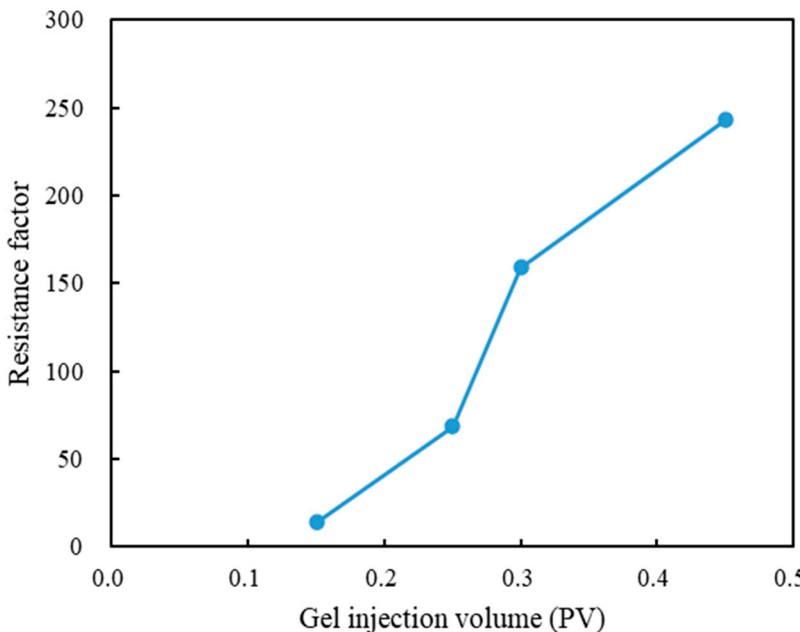

**Figure 5.** Curve of gel injection amount and resistance factor.

*3.6. Optimizing Total Injection Volume of Foam Slug*

In the experiment, three sand filling models were used to optimize the amount of foam injection, and the optimal amount of injection was 0.4, 0.7, and 0.9 PV. After the water drive of sand-filled core, foam liquid slug was injected, and then 0.3 PV of water was injected until the water drive pressure was stable. The water drive pressure was recorded, foam pressure was injected, and the resistance factor was calculated. Figure 6 shows the drawn curve. It can be seen that the relationship between the injected foam volume and the water drive resistance was that when the injected foam volume was 0.7 PV, the water drive resistance was the largest, the resistance factor was the highest, and the plugging effect was obvious. When injected with 0.4 PV of foam, the foaming amount was small, and the water drive resistance was small. When the amount of foam injected was 0.9 PV, the resistance to water acceleration was not as high as that of the injected foam, 0.7 PV.

This might be because the foaming agent component in the foam was adsorbed on the rock surface, forming a lubrication channel, which reduced the water drive resistance. Additionally, in the experiment, foam fluid flowed out of the core outlet, causing a waste of foam fluid. Accordingly, the optimal injected foam volume in the optimization room was 0.7 PV. In field implementation, pressure state, formation sealing, and channelling channel directionality should be considered, and the injection amount should be adjusted downward accordingly.

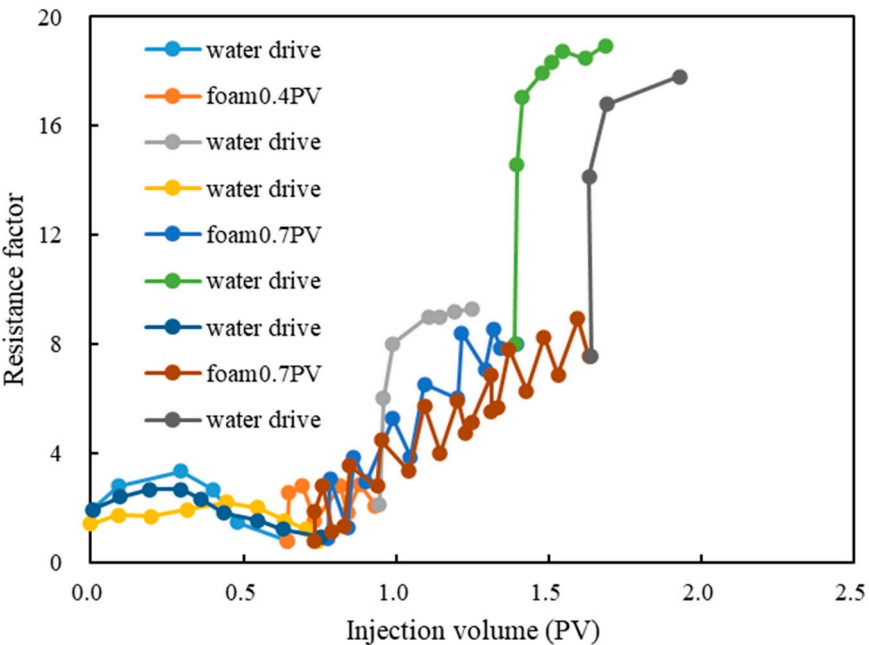

**Figure 6.** Curve of injected foam amount and resistance factor.

### 3.7. Indoor Optimization of Slug Injection Sequence

In the previous section, the basic parameters of separately injecting gel and foam fluid were optimized. In this section, gel and foam fluid slugs were injected together to optimize the plugging effect of foam fluid and gel injection sequence on heterogeneous formations. Effective program guidance was provided for field injection in the study area. Herein, three fractured cores with basically the same permeability and other properties were used. After water flooding to the water cut of 98%, three groups of experiments of foam and gel plugging were conducted, respectively. Table 3 illustrates the experimental scheme. During the experiment, 0.3 PV water was injected until the pressure became stable. The time was recorded, and Figure 7 shows a curve of displacement pressure and other parameters. After a water drive reached a high water cut, gel injection rapidly increased the water drive pressure to 230 times the water drive resistance, indicating that gel injection could quickly and effectively block the high permeability dominant channels such as large fractures. Scheme B: a foam liquid slug was injected after the water drive to high water cut, and then the gel slug. The water drive pressure rose to 280 times of water drive resistance after waiting for condensation, indicating that injecting foam liquid first can improve the water drive pressure after gel, which is conducive to plugging the high permeability layer. When the water drive reached a high water cut in Scheme C, foam liquid was injected immediately after the gel injection. The water drive pressure after gel also rose rapidly, which was 375 times the water drive resistance, significantly higher than the water drive pressure after gel in Schemes A and B. It shows that after a high water cut, the core first injects gel to block the large pore channel, and then injects foam liquid to further improve the plugging effect and reasonably optimize the profile control effect so that the water drive pressure can reach the optimal value. This is due to the gel system's initial solution into the complex fracture channel will choose the flow resistance is smaller

in the large aperture forward-pointing, in the large aperture to form the first blockage; due to the gel solution more show elasticity characteristics, in the process of large aperture flow encountered in the aperture of the smaller throat, it will produce the Jamin effect, resulting in a local pressure holding, forcing after the subsequent foam system liquid flow steering to take place into the next level of aperture flow, which helps to improve its wave volume and wash efficiency. It helps to improve its wave volume and oil-washing efficiency in complex fractures.

**Table 3.** Optimisation of injection sequence experimental scheme.

| No. | Fracture Permeability/$10^{-3}$ $\mu m^2$ | Injection Scheme | Subsequent Water Drive Volume |
|---|---|---|---|
| LE-A-3 | 233 | Scheme A: gel 0.3 PV | 0.3 PV |
| LE-A-5 | 206 | Scheme B: foam 0.6 PV + gel 0.3 PV | 0.5 PV |
| LE-A-6 | 237 | Scheme C: gel 0.3 PV + foam 0.6 PV | 0.5 PV |

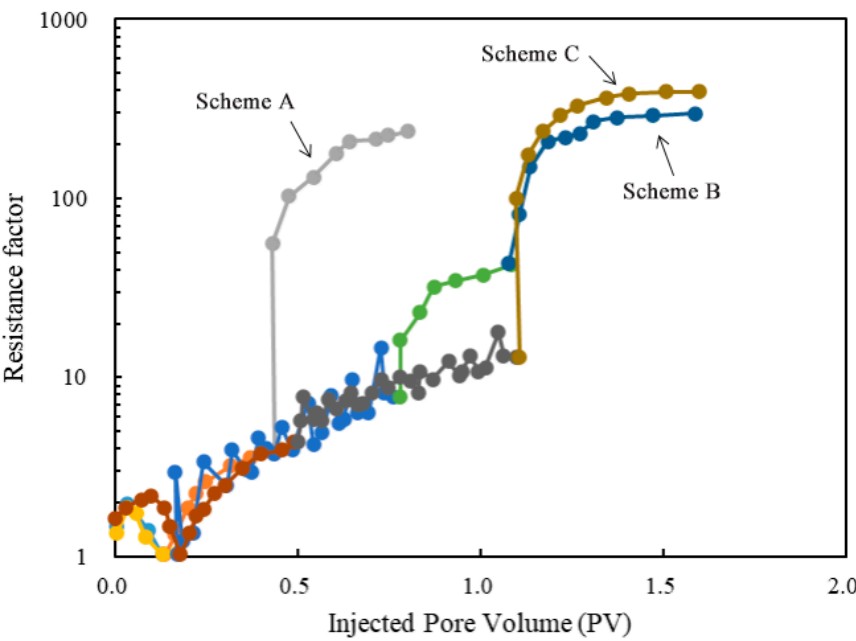

**Figure 7.** Resistance factor curves for different injection sequences.

*3.8. Double Tube Core Oil Displacement Efficiency Test*

Two cores with different permeability were displaced in parallel. The fractured core simulated the high permeability layer, and the sand-filling model simulated the low permeability layer to simulate reservoir heterogeneity. The experimental scheme was performed according to the optimized gel and foam injection parameters. Displacement pressure, liquid production, oil production, and other parameters were recorded. According to the injection scheme, after water flooding to 98% water content, 0.3 PV gel, and then 0.6 PV foam liquid were injected. After the gel was crosslinked, the injection pressure of water flooding increased rapidly and then maintained a stable increase, and the oil production gradually decreased after rising. The liquid and oil production of each slug were recorded, as shown in Figure 8. Herein, the final oil recovery of dual tube displacement reached 35.01%, with an increase of 23.69%. It shows that the injection of gel and small slug foam blocked the high permeability layer, making the injection profile even. Subsequent water drive produced a large amount of crude oil in the low-permeability layer, improving the recovery factor.

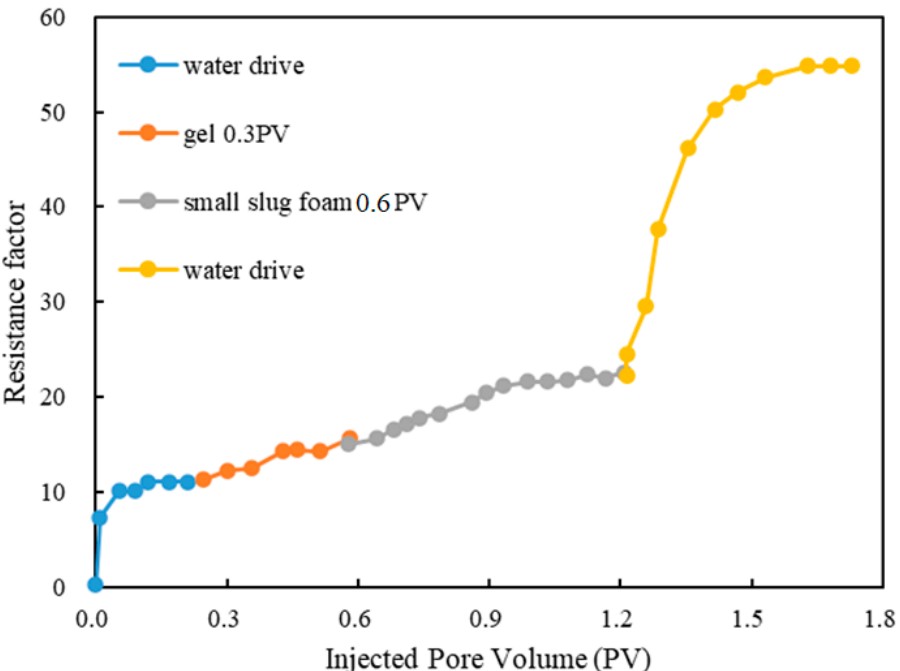

**Figure 8.** Oil recovery change curve of double tube displacement.

## 4. Oilfield Test

The optimized combined formula of gel and foam was applied in the oilfield of Baota in 2022 to improve the oil production in the tested block with an average permeability of $2.6 \times 10^{-3}$ $\mu m^2$. The total daily fluid output, daily oil output, and average water cut of the production wells correspond to the treated injection wells, Z93, Z92, and Z95 before the treatment were 110 $m^3$, 8.79 t, and 90.6%, respectively. The gel and foam were injected with the volumes listed in Table 4 per slug, and there were three slugs for gel and foam, respectively.

**Table 4.** Statistics of the injected fluid between the treatment.

| Well Name | Z93 | Z92 | Z95 |
|---|---|---|---|
| Volume of prepad fluid ($m^3$) | 25 | 28 | 32 |
| Volume per gel slug ($m^3$) | 100 | 110 | 125 |
| Volume per foam slug ($m^3$) | 200 | 220 | 250 |
| Volume of displacing fluid ($m^3$) | 25 | 28 | 32 |
| Bulk volume ($m^3$) | 950 | 1046 | 1189 |

The treatment was started at 1 November 2022 and the performance curves of the production wells were shown in Figure 9. It could be found form the curves that the fluid output decreased obviously, and the water cut also decreased by inches at the beginning of the treatment. Subsequently, the daily fluid output increased persistently, while the water cut continued to decreased, resulting a significant increase of daily oil output. From 1 March 2023 on, the performance became stable with more than 5.0 t of incremental daily oil output. By the end of June 2023, a cumulative oil output increase of 899 t and an average water cut decrease of 5% were acquired by the oil field test.

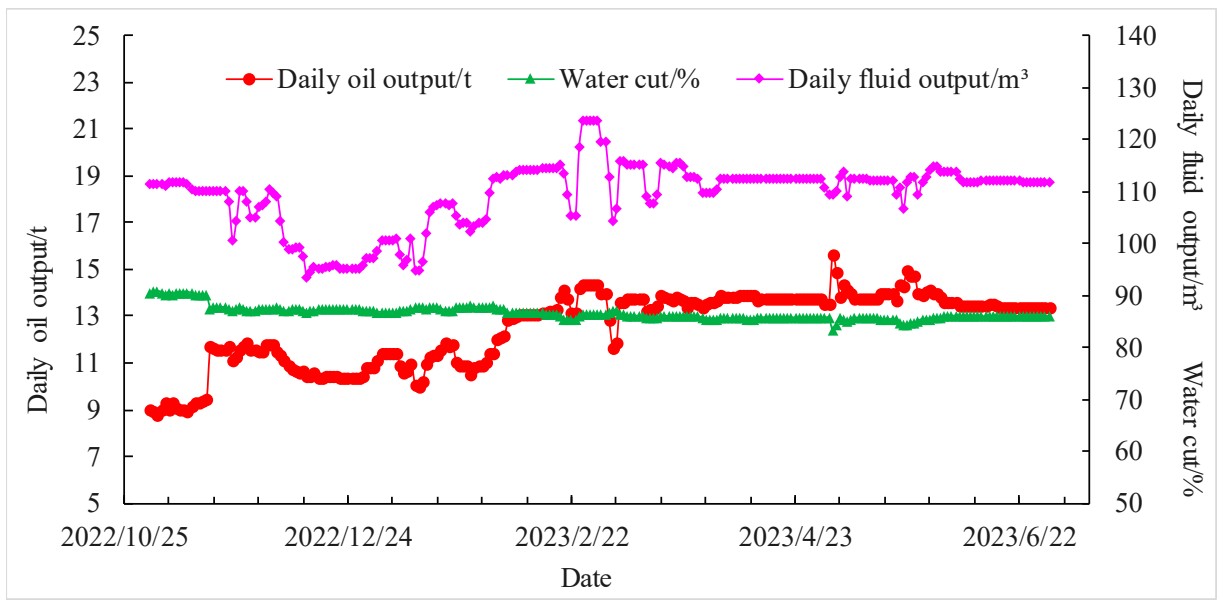

**Figure 9.** Effected oil wells production performance curve.

## 5. Conclusions

(1)    The nanoparticle composite foam was compounded with HDSX-1 and N1–2 through indoor optimization, the polymer was AP-P4, and the optimal injection volume of gel slug size was 0.3 PV.

(2)    In field implementation, pressure state, formation sealing, and channelling channel directionality should be considered, and the injection amount should be adjusted downward accordingly. Using the allowed equipment, we tried to use small slugs to inject foam liquid in multiple rounds, and the optimal total volume of injection in the room was 0.6 PV. After a high water cut, the core was first injected with gel to block the large pore channel and then with foam fluid to further improve the plugging effect.

(3)    The profile control effect could be further optimized reasonably, and the water drive pressure could reach the optimal value. Through the double-pipe indoor oil displacement experiment, the final oil recovery was 35.01%, with an increase of 23.69%. It shows that the injection of gel and small slug foam can achieve the purpose of plugging high permeability layers and even injection profiles.

(4)    Benefiting from the above schemes, there was an oil output increase of 899 t and an average water cut decrease of 5% acquired in the oil field test. Therefore, the optimized combination of gel and foam could significantly improve the recovery factor of heterogeneous reservoirs, which can provide strong guidance for designing a field injection scheme.

**Author Contributions:** X.G.: Conceptualization, Methodology, and Writing—Original Draft and Funding Acquisition. G.C.: Conceptualization, Methodology, Review, Editing, and Experiment. X.F.: Formal analysis, Methodology, and Experiment. Y.H.: Funding Acquisition and Experiment. F.H.: Formal Analysis and Validation. Z.G.: Formal analysis and Validation. S.K.: Language translation. All authors have read and agreed to the published version of the manuscript.

**Funding:** This research was funded by Natural Science Basic Research Plan in Shaanxi Province of China (2020JQ-787), Scientific Research Projects in Education Department of Shaanxi Provincial Government (20JK0829), and the National Natural Science Foundation of China (52104032, 52004216).

**Data Availability Statement:** The data presented in this study are available on request from the corresponding author. The data are not publicly available due to the research is still ongoing.

**Conflicts of Interest:** The authors declare no conflict of interest.

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
