# Peer review of "Evaluation of Foam Gel Compound Profile Control and Flooding Technology in Low-Permeability Reservoirs"

_processes, doi:10.3390/pr11082424_

Round 1

Reviewer 1 Report

The manuscript focuses on Evaluation of Foam Gel Compound Profile Control and Flooding Technology in Low Permeability Reservoirs, which benefits the developing efficiency of waterflooding oil recovery in fractured extra-low permeability reservoirs.

The principle of gel-foam composite profile control and flooding technology is to combine the advantages of mobile gel plugging of large channels and foam system plugging of small and medium channels. Compared with the single plugging technology, the composite technology has more advantages for oil increase and reduce water output in the high water cut period of fractured extra-low permeability reservoirs.The experiments, analyses, and conclusions are convincing. I recommend the manuscript to be published in processes after addressing the minor concerns listed below.

1) In table 1such as CTACBZ-1SD-AlSDlBK-2BK-7No. 1 foaming agentPO-FA330BK6AHDSX-1”,please mark the chemical content of these codes.

2) On page 9 "Indoor optimisation of slug injection sequence", please analyse in more detail why the injection sequence of gel first and then foam is the most appropriate.

3) On page 10, the sentences "0.3 PV gel and then 0.7 PV foam liquid were injected", please check the accuracy of "0.7 PV foam".

4) The current research conclusions are not very logical, please mark them as classified and written according to (1), (2) and (3).

5) As the author is not writing in native language, please check the sentences in the paper for readability and accuracy.

The grammar of this manuscript should be revised to improve its readability.

Author Response

Response

1)A description of the chemicals highlighted in red has been added to Table 1 below.

2)“This is due to the gel system initial solution into the complex fracture channel will choose the flow resistance is smaller in the large aperture forward pointing, in the large aperture to form the first blockage; due to the gel solution more show elasticity characteristics, in the process of large aperture flow encountered in the aperture of the smaller throat, it will produce the Jamin effect, resulting in a local pressure holding, forcing after the subsequent foam system liquid flow steering to take place into the next level of aperture flow, which helps to improve its wave volume and wash efficiency. It helps to improve its wave volume and oil washing efficiency in complex fractures.”We added the above explanation on page 10 of the manuscript of the paper.

3)We would like to thank the reviewers for their careful discovery that this category should in fact be 0.6 PV foam, which we have corrected in the manuscript.

4)We've revised the conclusion as requested by the reviewer and highlighted it in red.

5)We have checked the accuracy of the language used in the paper and have highlighted it in red.

Reviewer 2 Report

The topic of this paper means a lot for the development of the oil reservoirs with low permeability and water channeling. The contents and the results show credible and logical. I suggest to publish this paper after a minor revision as following.

a.  Supply the parameters such as temperature and salinity in the experiments.

b.  Please add the related contents, if the conclusion resulted form this paper were applied in the oilfield.

c.  There are some linguistic problems, please refine the writing.

Author Response

a.The value of the parameters in the experiments were supplied in the section of foaming agent optimization. The values such as temperature, salinity and PH

were selected according to the formation of the studied Baota oil field.

b.The contents of the oil field application were supplied in the section of oilfield test. The optimized combination of the gel and foam were applied in three injection wells. An oil output increase of 899 t and an average water cut decrease of 5% were acquired from the correspond prodction wells

c.We have checked the accuracy of the language used in the paper and have highlighted it in red.
